# Marine-Derived Polymeric Materials and Biomimetics: An Overview

**DOI:** 10.3390/polym12051002

**Published:** 2020-04-26

**Authors:** Marion Claverie, Colin McReynolds, Arnaud Petitpas, Martin Thomas, Susana C. M. Fernandes

**Affiliations:** 1E2S UPPA, CNRS, IPREM, Universite de Pau et des Pays de l’Adour, 64600 Anglet, France; marion.claverie@univ-pau.fr (M.C.); c.mc-reynolds@univ-pau.fr (C.M.); petitpas.arnaud@outlook.fr (A.P.); martin.thomas@univ-pau.fr (M.T.); 2Department of Chemistry—Angstrom Laboratory, Polymer Chemistry, Uppsala University, Lagerhyddsvagen 1, 75120 Uppsala, Sweden

**Keywords:** Blue biotechnology, by-products valorization, marine polysaccharides, marine proteins, marine secondary metabolites, application sectors, bio-inspiration

## Abstract

The review covers recent literature on the ocean as both a source of biotechnological tools and as a source of bio-inspired materials. The emphasis is on marine biomacromolecules namely hyaluronic acid, chitin and chitosan, peptides, collagen, enzymes, polysaccharides from algae, and secondary metabolites like mycosporines. Their specific biological, physicochemical and structural properties together with relevant applications in biocomposite materials have been included. Additionally, it refers to the marine organisms as source of inspiration for the design and development of sustainable and functional (bio)materials. Marine biological functions that mimic reef fish mucus, marine adhesives and structural colouration are explained.

## 1. Introduction

Representing around 70% of the biosphere and encompassing 34 of the 36 phyla that categorize living things, marine biodiversity is extremely vast [1]. Life appeared in the oceans more than 3.6 billion years ago and ever-changing conditions (temperature, salinity, tides, pressure, radiation, light, etc.) have forced species to evolve and develop extraordinary physical and chemical strategies to exploit diverse ecological niches. This diversity has driven inspiration for research over the decades.

Blue biotechnology, referring to the exploration and exploitation of the resulting diverse marine organisms in order to develop new products is a constantly expanding field, with important implications for sustainable use of biological oceanic resources [2]. A current approach is that of biomimetism, that is to say imitating natural functions and structures, using marine organisms as source of inspiration. In this way scientists have developed materials with numerous applications in human health, agriculture, energy, aquaculture, fine chemicals, pharmaceuticals, food, cosmetics, and environmental sectors including biosensing and bioremediation [3,4,5]. Biomimetic approaches are gaining interest and the number of studies based on marine-derived or marine-inspired materials is constantly growing. The present review presents an overview of some important recent advances in this field.

Few reviews are focused on the ocean as both a source of biotechnological tools and as a source of bio-inspired materials. In this review, the reader will find compilations on marine biomacromolecules and their specific properties together with relevant applications in biocomposite materials. Some studies demonstrating how marine organisms may be sources of inspiration for the design and development of sustainable and functional (bio)materials are also presented.

## 2. The Ocean as a Source of Biotechnological Tools

Marine biodiversity represents a wide reservoir of molecules with an extraordinary diversity of chemical functions, structures and architectures. Marine molecules extracted from animal, algal or microbial sources display a full spectrum of properties and are used in multiple domains of applications. From small bioactive molecules (mainly secondary metabolites) to large biomacromolecules (proteins and polysaccharides), marine natural products have been increasingly used in cosmetics and in medicine to treat cancer and chronic diseases but also in cell therapy and tissue engineering [1,6,7,8]. Moreover, marine polymers are especially promising alternatives to synthetic polymers for the development of sustainable and eco-friendly materials, including bioplastics and biocomposites [8].

Herein, some of the best-known and most used marine biomacromolecules and secondary metabolites are covered and their main applications are described.

### 2.1. Marine Biomacromolecules

Biomacromolecules, will be defined here as large organic molecules fulfilling diverse structural or biological roles in living organisms.

In this case, a prime example is polysaccharides, long chains of individual sugar units linked through glycosidic bonds that present different structure and biological function. Marine organisms, crustaceans, cnidarians and diverse species of fish among the Animalia, as well as algae, are well-exploited sources of polysaccharides, each with unique physico-chemical, mechanical, morphological and biological properties that suit for very diverse applications.

Another example of biomacromolecules is proteins, which play many critical roles in the body. Proteins are the most abundant organic compound in animals. They are made of large and complex molecules that form polypeptides and ultimately proteins. Marine proteins and peptides can be extracted from diverse marine organisms including fish, shellfish, algae, crustaceans, mollusks, echinoderms and sponges [9]. Indeed, depending on their 3D structures and amino acid sequences, proteins and peptides from marine origin exhibit a wide range of biological functions including antioxidant, antimicrobial, anticancer, immunomodulatory, antihypertensive, anti-ageing, anticoagulant and anti-diabetic effects [10].

In the following sections, relevant animal- and macroalgae-derived biomacromolecules will be described. Among the animal-derived molecules, glycosaminoglycans (GAGs) and chitin are well-studied and are presently used in various domains. Accordingly, some emphasis will be on hyaluronic acid (HA—a type of glycosaminoglycan) and chitin-derived polymers. Protein-based compounds can comprise a high percentage of the mass of marine species: proteins, peptides derivatives and enzymes – opening a wide range of potential applications across many sectors. We also will suggest the potential of marine proteins or peptides (with a particular focus on collagen) and marine enzymes in biomaterial domain.

Secondly, macroalgal biopolymers will be described. Like the above, these compounds have been the subject of research and dedicated industrial exploitation for over a century. Their properties make them particularly amenable to new modification and bio-inspired use.

#### 2.1.1. Polysaccharides from Marine Animals

● Hyaluronic Acid

Hyaluronic acid (HA, Figure 1), also called hyaluronan, is a non-sulfated GAG. It is a linear polysaccharide that assures important biological functions in bacteria and mammalian tissues [11]. In terms of structure, HA is composed of repeating disaccharide units linked by β-(1,4) and β-(1,3) bonds. Its molecular weight (MW) depends on its origin in the organism, varying from 5 to 20,000 kg/mol. It is synthesized by transmembrane HA synthases and degraded by hyaluronidases.

The most common sources of HA are connective tissues containing synovial fluid, eyeball vitreous fluid, umbilical cords and rooster combs [11,12,13]. Nonetheless, the potential of marine organisms as sources of HA has received increasing attention because of the risks of animal-derived pathogens and inter-species viral contamination of conventional HA sources [13,14]. In marine organisms, it is possible to extract HA from stingray livers, bivalves, and fish cartilage matrix and eyeball vitreous humor, the latter being optimal [15,16]. Even if it is an important structural element of the aggrecan in cartilaginous fishes, low content renders it economically inaccessible via current industrial extraction practices [17].

The structure and properties of GAGs derived from marine animals, and in particular their MW and sulfatation state, are different from those derived from terrestrial organisms. These aspects influence their ability to interact with proteins and are the principal factor determining therapeutic properties [18]. For HA, size has been shown to be the principal factor that governs its physico-chemical properties [19]. The tight control of HA structure and concentration plays a role in cell proliferation and migration but also in angiogenesis and inflammation [20].

HA is used principally in the biomedical sector for its viscoelastic, hygroscopic, lubricating and water-retaining properties, and is largely biocompatible. Indeed, even at low concentration (1%), HA is able to form hydrogels. These gels are usable alone or can be used as a vehicle for long term, low-dose drug delivery [21]. HA is actively used for drug and growth factor delivery, or as scaffolding material to stimulate bone healing and biomineralization [22]. Notably, osteoarthritis has been treated with HA for more than 40 years [19].

Thanks to its high biocompatibility, HA is a promising polymer for the development of novel biomimetic materials. For instance, the design of fibre scaffolds mimicking soft-tissue mechanical properties and tunable porosity, for materials that stimulate wound healing [23]. Three dimensional (3D) HA hydrogels can be used as biomimetic culture systems to investigate the behaviour of glioblastoma stem cells according to microenvironmental conditions and regulation, to propose novel therapeutic strategies [24]. Furthermore, HA is an essential element for novel biomimetic bioelectrodes. These are artificial conductive materials made of (HA)-doped polypyrrole films and are capable of efficient electrical signalling and preferential biological interactions depending on the molecular weight of the HA used during the electrochemical synthesis. These could be used for neural recording and neuromodulation [25].

The chemical or enzymatic addition of functional groups is a way to enhance HA mechanical strength and control its biodegradation making it tunable for use as scaffold in regenerative medicine [20,26]. A large variety of HA/HA-derivatives formulations are accessible and their curative effects mostly depend on their rheological properties and administration routes [19].

● Chitin and Chitosan

Chitin was first isolated from mushrooms by the French botanist H. Braconnot in 1811. It is one of the most abundant polysaccharides in nature and the presence of nitrogen in its chemical structure distinguishes it from other sugars [27]. This structural biopolymer is found in a vast number of organisms (animals, algae and fungi) under the form of ordered crystalline microfibrils. It acts as a supportive and protective component and is the main component of the exoskeleton of insects, mollusks and crustaceans. Due to its inherent insoluble nature, chitin was largely ignored as a resource until recently [28]. Currently, chitin is extracted at industrial or semi-industrial scale from shrimp, crab and lobster shells, easily available and abundant by-products of the shellfish processing industry [29]. It is biodegradable, although it does inhibit some microbial activity, and biocompatible displaying low cytotoxicity and immunogenicity.

Structurally, this high-molecular-weight linear polysaccharide is composed of N-acetyl-2-amido-2-deoxy-d-glucose units linked by β(1→4) bonds (Figure 1). Isolated chitin is a highly ordered polymer in which N-acetyl-d-glucosamine is in higher proportion than d-glucosamine. During its biosynthesis, the arrangement of the individual chitin chains results in chitin nanofibrils that further aggregate into nanofibres and finally chitin nanofibres bundles. Over the last 20 years, research on chitin has principally been focused on its nanocrystal or nanofibres forms. These chitin nanoforms can be used as reinforcing agents in numerous applications like nanostructured materials [28].

The most important derivative of chitin is chitosan, resulting from its deacetylation. This process, performed by acidic or enzymatic methods, cleaves the N-acetyl group at C-2 position of sugar units and results in a chain of 2-amino-2-deoxy-d-glucose units linked through β(1→4) bonds (Figure 1). Chitosan is also found in native form in the cell walls of some fungi [27]. Depending on the origin and the processing methods used to treat the raw material, chitosan structure and characteristics (purity, N-acetylation degree and molecular weight) are tunable [27,30]. The greatest advantage of chitosan compared to chitin is its solubility. Indeed, thanks to its free amino groups, chitosan can be solubilized in dilute acidic aqueous media while dissolving chitin is much more challenging. Once solubilized, chitosan can readily be transformed into hydrogels, 3D porous scaffolds, membranes and films displaying interesting mechanical strength and permeability properties [31]. Furthermore, being a polycation, chitosan is very distinct from other polysaccharides that generally are anionic or neutral. As such, chitosan is easily modifiable using chemical or enzymatic pathways. The functionalization of its amino groups allows amidation, quaternization, alkylation (through reductive amination), grafting and chelation of metals, like copper and silver, for instance. On the other hand, hydroxyl groups of the monomer units are also available sites for esterification, *O*-acetylation, grafting and H-bonding [32].

The different possibilities to modify chitosan characteristics and improve its solubility, biocompatibility as well as its adhesive, antibacterial, antifungal, antiviral and non-allergenic properties have resulted in the development of many new advanced materials [33,34,35,36,37]. Moreover, chitosan is non-toxic and can be biodegraded by lysozyme contained in human body fluids [38,39]. Owing to all these characteristics, chitosan is actively investigated and presents great potential in a multitude of applications including tissue engineering, drug delivery, wound dressing, scaffolds, pharmaceutical contaminant removal, cancer diagnosis, (nano)composites, high-tech materials, food, packaging, dye removal, and more [33,35,37,40,41,42,43,44,45,46,47,48,49,50,51,52,53,54].

Overall, chitin and its derivatives represent promising matrices for the development of novel materials through biomimetical approaches [34,55] and notably in the recent field of “extreme biomimetics” for the development of nanoscale-structured composites [56,57,58]. In this field, chitin isolated from marine sponges is of high interest for the synthesis of bioinorganic composite materials made under high temperatures and pressures. These novel biomaterials display electrical, chemical, and material properties and could be applied in water filtration, medicine, catalysis and biosensing [58].

#### 2.1.2. Marine Proteins

● Marine Collagen, Proteins and Peptides

Among animal-derived marine protein biomacromolecules, collagen is the most abundant protein in human and animal bodies, and perhaps the most widely used pure protein sourced from marine organisms [59]. This polypeptide is the major component of bone, skin, cartilage, and tendon, where it is organized into fibrils sheets by fibroblasts [60]. Marine collagen is mainly extracted from the skin, bones and scales of fish, or from jellyfish, sea urchin, starfish or sea cucumber connective tissue. Different collagen subtypes exist and differ from each other in structure and properties, which influences cell signalling and adhesion functions [61]. Marine collagen is almost exclusively type 1 collagen, and has generally MW as compared to other sources. This makes it more bioavailable and absorbable. Thanks to the removal of all oils during processing, marine collagen has no smell or taste.

A growing number of biomimetic composites based on collagen from marine origin are being developed and display very promising properties [62,63,64]. Marine collagen contains high amounts of hydroxyproline, an amino acid essential for skin, blood vessels and other connective tissues, which makes it very attractive for a variety of uses in cosmetics, functional food and tissue engineering [62]. As opposed to mammalian collagen, which poses pathological risks in the transmission of viral diseases, marine collagen, is considered safer and easier to extract. Indeed, marine-derived collagens were proved to be less immunogenic and may avoid secondary complications occurring when used in human body [65]. Marine collagen is known to increase skin moisture levels and help protect against the UV exposure and photoaging effects and therefore is of high interest for use in cosmetics. Marine fish collagen is also used in various biomedical applications as drug and gene delivery systems but also as scaffolding in stem cells based regenerative medicine. An emerging sector concerns its use in bone tissue engineering through the development of bone grafting materials consisting of collagen composites or scaffolds. In that field, in vitro osteogenesis may be promoted by a porous biomimetic environment based on the structure and chemical composition of collagen-containing marine sponges and mimicking the cancellous architecture with the complex canal system of bone tissue [63].

Marine proteins and peptides other than collagen are also of interest due to their broad spectrum of biological activity. Use and research of marine proteins and peptides is increasing across multiple industrial sectors including foods, nutraceuticals, pharmaceuticals, and cosmeceuticals [66]. In the pharmaceutical industry, marine proteins are being explored for the treatment or prevention of various diseases [67]. Recently, fish proteins and peptides derived from chemical and enzymatic hydrolysis were used for the development of novel antioxidant cosmeceuticals [66].

● Marine Enzymes

Although investigation into the diversity of marine biocatalysts is lagging behind terrestrial sources, in recent years the marine environment has gained increasing importance as a source of novel enzymes [2].

Marine enzymes including proteases, peroxidases, oxidoreductases, hydrolases, transferases, isomerases, ligases and lyases are already produced for food, industrial and medical applications [68,69,70].

Thanks to recent progress in metabolomics, biocatalysts are not only extractable directly from marine organisms but may also be produced through heterologous expression in marine hosts, such as cyanobacteria or microalgae, that are cultivatable for industrial applications [71,72,73]. In this review, we will focus on marine biocatalysts as tools used for the extraction of marine molecules and post-extraction molecular modification. Enzymes are efficient in mild reaction conditions and are very specific, enhancing accessibility of biomolecules and allowing for their functionalization through a more eco-friendly process compared to purely chemical-based methods [74]. For instance, enzyme-assisted extraction can be used to improve the recovery of specific bioactive compounds from marine resources like seaweeds [75,76] and shrimp shells [77], for example. Furthermore, a multitude of marine biocatalysts can be used for structural and chemical modifications of polymers. As such, marine enzymes are efficient tools for the treatment of raw materials, such as marine polysaccharides to design novel biomaterials. The properties of polysaccharides are very size-dependent and short chains are often desired. For example, the use of fucoidans as pharmaceutical products requires the preparation of oligomeric products strictly homogenous in size. To that end, fucoidanases from marine bacteria [78] or molluscs [79], that are able to depolymerize high MW fucoidans, are ideal for the production of defined fuco-oligosaccharides [78,80]. By degrading alginate, alginate lyases are also highly efficient tools to prepare oligosaccharides [81,82]. Compared to complete, high MW alginate, alginate oligosaccharides are more bioavailable while maintaining the physiological functions and activities of the polysaccharide [83]. Concerning marine collagens, collagenase from marine bacteria may be used for the synthesis of collagen hydrolysates with improved antioxidant properties [84].

The use of marine enzymes for biotechnological applications is growing quickly and the potential of marine biocatalysts, many of which remaining undiscovered, is immense.

#### 2.1.3. Biopolymers from Macroalgae

Macroalgae, commonly known as seaweeds are macroscopic, multicellular algae, which include three broad and distinct evolutionary groups: the rhodophyta (red seaweeds), the phaeophyceae (brown) and the chlorophyceae (green). Within this biological diversity lies considerable chemical diversity, with characteristic polymers within each group. Exhaustive review of these polymers is out of the scope of this review; however, the main types of polymers that these organisms produce are briefly summarized.

Polymers of interest among these organisms are their characteristic structural and non-structural polysaccharides (Table 1 adapted from comprehensive review by [85]), which can represent over 70% of dry weight [86]. Some are extensively extracted for industrial uses, particularly the agars and carrageenans in red seaweeds, alginates in brown seaweeds being of particular industrial importance as hydrogels [87].

With the exception of storage polysaccharides (akin to starch in plants) they are located in or associated with the cell wall. Cellulose is the most abundant in terms of occurrence across the different taxa, although physical properties (eg., microfibril diameter) and the enzymes involved in their synthesis vary [88]. In seaweeds, matrix polysaccharides are almost universally sulfated, a feature that does not have an equivalent in terrestrial plants. Sulfatation is particularly important in the biological activity of these biopolymers [89]—in depth study of the bioactivity of these biopolymers may pave the way towards tailored molecules with high therapeutic potential [90,91,92].

From a biomimetic standpoint, bringing together the biological roles of these biopolymers, and their chemical characteristics (structure, derivatisation with small molecules) is of high interest. As they provide mechanical support for the structure of microalgae, it is possible to use them for materials that retain the key features: matrix mechanics and permeability; the ability to sequester and to deliver drugs, proteins, and or nucleic acids; provide receptor- or enzyme-mediated cell-matrix interactions [94]. Existing examples include the incorporation of specific nanofibrils of cellulose in polysaccharides matrices provides adaptive viscoelastic behaviour and results in tissue that is stronger, more extensible and more fatigue resistant [95,96]. Calcareous coralline algae were recently shown to produce chitin and collagen-like proteins, with the two biopolymers likely playing a role in the biomineralization process of the algal skeletons [93,97]. Reproducing naturally-occurring combinations of these biopolymers can result in materials with tunable characteristics, i.e., hydrophobicity, rigidness, mechanical resistance, etc. [98].

### 2.2. Marine Secondary Metabolites

Also known as marine natural products, marine secondary metabolites are small organic molecules that play an important role in the ecology of marine organisms—as chemical defences, reproductive signalling and more. Many thousands of compounds with potential antibiotic, cytotoxic and diverse medical and industrial use have been described from the marine environment and are produced by organisms as secondary metabolites.

Marine invertebrates like sponges, bryozoans, tunicates, mollusks, marine cyanobacteria and macroalgae represent largest and most investigated chemical diversity of marine natural products.

In this section, special attention is paid to the marine secondary metabolites that are mycosporines and mycosporines-like amino acids (altogether abbreviated here as MAAs).

#### Mycosporines and Mycosporine-Like Amino Acids

To deal with UV exposure, marine organisms have evolved complex DNA repair mechanisms as well as biosynthesis and accumulation of UVR absorbing molecules such as MAAs.

MAAs are small molecules with MW generally lower than 400 Da (Figure 2). They are Schiff bases (enamino ketones) are divided into two main groups: the aminocyclohexenones (mycosporines) and the aminocyclohexenimines (mycosporine-like amino acid) substituted with a nitrogen atom and linked with amino acid or amino alcohol (Figure 2) [99,100]. MAAs are highly water soluble zwitterions with diverse functional groups. Over 30 different chemical structures of MAAs are currently described [101].

These UV absorbing molecules have been identified in various taxonomically diverse organisms that includes marine heterotrophic bacteria and fungi (*Zygomycetes*, *Deuteromycetes*, *Ascomycetes*, *Basidiomycetes*, *Bacillariophyta* and *Aphyllophorales*) [102,103,104,105], cyanobacteria (blue green algae, prokaryotes) [106,107], about 152 species (206 strains) of microalgae [108,109,110], macroalgae [111], marine fishes (eyes, skin, tissues) [112], phytoplankton, echinoderms (*Artemia*, *Anaspidea*), marine molluscs (*Ascidiacea*) [111] and many coral reef species (*Zoantharia* symbiotes of *Scleractinia* corals and *Tridacna* clams, *Scyphozoan* jellies) [112].

Biosynthesis of these molecules is still not perfectly known, but there are two described pathways with associated genetic information: the shikimate pathway and the pentose phosphate pathway [99].

● Properties of MAAs

MAAs are well known for their photo-protective properties: high UV absorption, high molar extinction coefficient, antioxidant activity and high photo-stability. These properties are dependent of the different side groups and nitrogen substituents [113].

A part to be recognized as sunscreens due to their absorbance properties, in nature, these small active molecules may also be involved in many other biological processes, potentially acting as an intracellular nitrogen reservoir, DNA-protection, osmotic regulation and they also take part in cellular interactions [114,115,116].

In humans, MAAs have anti-inflammatory properties, and are shown to increase fibroblast and keratinocyte multiplication, among other cellular activities [116].

● MAAs as Biotechnological Tools


*MAAs as Sunscreen Compounds*


In nature, UV light induces the biosynthesis of these MAAs in certain marine organisms, to function as sunscreen compounds. They exhibit maximum absorbance in the ultraviolet spectrum, i.e., in the UV-A (315 nm–400 nm) and UV-B (280–315 nm) range and their molar extinction coefficient (ε) ranges from 28,100 to 50,000 M^−1^ cm^−1^ [100,116]. The photo-excited states of MAAs have been shown to relax in a controlled dissipation of the energy as heat without the generation of reactive oxygen species (ROS) thus preventing oxidative stress [99,100,116]. These compounds thus have great potential for use in UV-protective cosmetic products.

Photodegradation, photophysical studies and thermoresponsive studies clearly indicates that these MAAs are both photostable and thermally stable sun-sunscreen compounds [117].

These molecules have an edge over conventional sunscreens, which may have detrimental effects in marine ecosystems, have low photostability, and are easily degraded into, or generate free radicals [118,119,120].

Experimental studies based on irradiation of A375 human melanoma cells, fibroblast cells IMR-90 [121] and HaCaT cells [122] were conducted with UV rays in the presence of different MAAs, namely glutamine, porphyra-334 and palythine, respectively. The in vivo assay results revealed that MAAs had a protective effect against UV-induced DNA damage.

With these advantages, MAA-based products have started to be commercialized. Two European companies sell sunscreen ingredients containing the MAA shinorine extracted from *Porphyra umbilicalis*, a type of red algae [123]. Osborn et al. [124] from the Oregon State University have found that gadusol (MAA precursor that also provides protection against UV-B) was also synthesized by fishes. Based on their researches they have launched a company, Gadusol Laboratories, to engineer sunscreens based on gadusol [123,124]. Paul Long’s group, in King’s College London, use palythine, extracted from red algae *Chondrus yendoi,* for protection against both UV-A and UV-B. They filed a patent application for this molecule and licensed it to a cosmetic company in London [125].

MAAs could also be used as photo-stabilisers for the protection of plastics, paints and varnishes [126].


*MAAs as Antioxidants*


Reactive oxygen species (ROS) or free radicals are produced during the oxidation step for energy metabolism in most biological process, these species activate cellular mechanisms such as cell division, inflammation and stress responses. ROS generation and scavenging are in equilibrium in these mechanisms. Overproduction of ROS can result in deleterious physiological effects, such as inflammation and protein oxidation [116].

Most MAAs have the ability to scavenge free radicals (ROS) such as singlet oxygen, superoxide anions, hydroperoxyl radicals, and hydroxyl radicals due to UV radiation or environmental stress [116]. The antioxidant property of these molecules is attributed to the presence of a cetonic group (Figure 2) [127]. The slow photodegradation of MAAs with the reaction of singlet oxygen and in the presence of a photosensitizer are indicative of their antioxidant activity [128].

### 2.3. Marine Molecules as Tools for Biocomposite Materials and Their Applications

Recently, breakthroughs have been made in the development of biocomposite materials, with growing awareness and adoption of eco-friendly materials from sustainable resources. In some sectors, such as the automotive industry, the low weight and environmental impact of these biocomposite materials are very desirable.

In light of this interest, researchers have been paying considerable attention to marine-derived biomacromolecules and biomolecules with properties appropriate for various applications [129,130,131,132,133,134]. Since the usefulness of biopolymers taken alone is limited, biocomposites combining the characteristics of two or more marine biomaterials are of growing interest. This strategy can result in new hybrid materials with better properties than those used separately. Some of the most promising areas for the application of these biocomposites are in the medical and food packaging industries. However, they are also noteworthy for their mechanical properties for materials in diverse sectors, and in wastewater treatment [129,130,131].

In this section, the current trends in marine biomacromolecule-derived biocomposites and the wide variety of marine-based materials available for those applications are presented.

#### 2.3.1. Marine-Based Biocomposites for Food Packaging

The food packaging industry is responsible for a large amount of polluting plastic waste, thus environmentally friendly alternatives are becoming urgent. Marine polysaccharide-based biocomposites present some required properties for the design of sustainable food packaging materials, with their biodegradability, biocompatibility, low toxicity and renewability [135]. Predominantly derived from polysaccharides with relatively poor mechanical performance, reinforcement with other bio-sourced polymers is often necessary, particularly nanocrystals or nanofibres.

In this way, alginates reinforced with cellulose nanocrystals extracted from birch pulp were developed by Sirvio et al. [135]. Their final product presents high mechanical strength and excellent grease barrier properties as well as reduced water vapour permeability—essential characteristics for application in food packaging.

Going further, an ideal food packaging material would also interact with the food to improve preservation. In this regard, chitosan is a good candidate: in addition to the properties discussed above, chitosan possesses antioxidant and antimicrobial properties. Chitosan-based biocomposites can be sustainable, recyclable, and biodegradable materials, particularly when reinforced with nanofillers to improve mechanical resistance and water-/gas-barrier properties. Packaging products able to actively protect the food also reduce food waste, which is a significant problem nowadays. This is currently actively investigated as it is the goal of the M-era.net collaborative project BIOFOODPACK, launched in 2016 [136].

An example of this active preservation is the innovative chitosan-genipin films designed by Nunes et al. [137]. The films were developed to replace the sulfur dioxide as preservative during the process of making white wine. The chitosan-genipin films were able to inhibit oxidation reactions as well as microbial growth while not affecting the organoleptic properties of the wine, making them a possible sustainable and efficient alternative method for wine conservation, with economic and environmental advantages.

#### 2.3.2. Marine Nanofillers for Structural Applications

The natural fibre biocomposites market is experiencing considerable growth, especially due to diverse applications and the expansion of the eco-design concept in the aerospace and automotive industries. When combined with a biodegradable matrix, natural fibres (and plant fibres, in particular) can offer a new perspective on product lifecycle, with noticeable consequences on their environmental impact. For example, Pervaiz and Sain (2002) estimated that the replacement of 50% of glass fibre composites with hemp-based biocomposites could potentially reduce annual carbon dioxide emissions by 3 million tons, at the same time saving around 1 million m^3^ of crude oil in the process—for the automotive industry alone [138].

While focus has been mainly on terrestrial plants, natural fibres harvested from the marine resources in structural applications seem to be overlooked at the moment.

Nevertheless, the oceans are be the source of fibres with excellent mechanical properties. The sea grass *Zostera marina* (harvested from the Baltic coast) is particularly suitable for the development of biodegradable structures with good stiffness and low density [139]. Its microfibres (5 μm in diameter) are mainly composed of cellulose (~55%), polysaccharides like xylan (~40%) and Klason lignin (~5%). Its mechanical tensile properties are quite similar to other terrestrial plant fibres such as jute and sisal, making it a good candidate for biosourced reinforced eco-friendly composites. Sea grass is locally abundant and large quantities may wash ashore on beaches. Indeed, what is currently a costly nuisance could become a good sustainable source of natural fibres for industrial applications [139].

It should be noted that the use of the great mechanical properties of sea grass is not recent. Indeed, 8000 years ago ancient civilizations were already using sea grass to make cordage and ropes as evidenced by vestiges discovered during the excavation archaeological sites in coastal California [140].

Despite not having the required mechanical properties for structural applications by itself, polysaccharides like alginate derived from seaweed can be used as a binder for wood fibres/textile waste fibre composites [141]. These biocomposites are light and semi-rigid, with very low thermal conductivity (around 0.08 W/m/K), and as such have potential applications as both insulation and for structural purposes.

Pure chitin in nanofibre and nanocrystal forms has recently gained interest for use in nanostructured biocomposite materials [28,142,143,144]. These nanofillers combine the unique physicochemical, mechanical and biological properties of native chitin and the intrinsic characteristics of the nano-size materials like low density and high aspect ratio. In 2001 Paillet and Dufresne used chitin nanocrystals for the first time as ecofriendly reinforcing fillers in thermoplastic nanocomposites [145]. In the following years, new domains of exploitation of chitin nanofillers have emerged including bionanocomposite materials, electronics and medical devices [43,146,147,148,149,150,151].

#### 2.3.3. Marine-Based Biocomposites for Water Treatment

Water pollution caused by industrial activities is one of the major concerns of this century. Various methods have been developed to cleanse wastewaters in order to obtain clean and usable water, a vital resource. Amongst them, adsorption, a process in which the pollutants are collected from the water without the production of harmful by-products, is very popular due to its simplicity, convenience, cost-efficiency, and the availability of sustainable biosourced-based adsorbents [152]. Given their abundance, non-toxicity and large number of functional groups, marine polysaccharide-based adsorbents are good candidates for the removal of heavy metal ions and dyes from wastewaters.

Alginate- and chitosan-based hydrogels are some of the more widely studied bio-adsorbents for such application [153,154,155,156,157]. Indeed, as three-dimensional cross-linked polymer networks, hydrogels can absorb large amounts of water, increasing the possibilities and efficiency of interactions between the functional groups of the adsorbent and the targeted soluble substances.

It has been shown that alginate-based cross-linked beads are able to remove relatively high amounts of toxic metal ions and cationic dyes [153]. In particular, this biomaterial possesses ultrahigh adsorption capacity for methylene blue and methylene violet dyes (respectively, 2977 and 2105 mg/g) as well as for Pb^2+^ ions with a maximum capacity of 2042 mg/g. An alginate-cellulose double-network frame composite was also elaborated by Zhan et al. to target the removal of Cu^2+^, Zn^2+^ and Pb^2+^, with some success: the material had an adsorption capacity of 177.1, 110.2 and 234.2 mg/g, respectively [154].

As stated previously, chitosan is widely investigated for the development of bio-adsorbents. For instance, chitosan beads modified with 3-aminopropyl triethoxysilane resulted in significant increase of the adsorption maximum capacity, from 317.23 mg/g to 433.77 mg/g of reactive blue 4 dye (as compared to untreated chitosan beads) [155]. Chitosan can also be paired with natural fibres, bringing together favourable mechanical properties and functional characteristics in order to improve overall performance of the adsorbent [156,157]. This is the case for the chitosan-based biocomposites film reinforced with water hyacinth fibres recently developed by Pisitsak et al. [157]. The addition of bio-fillers increased the adsorption capacity of Cu^2+^ ions from 6.4 mg/g to 34.1 mg/g, while improving the tensile modulus of the film by 400%. Further, Labidi et al. 2019, studied the equilibrium, kinetic, thermodynamics and regeneration of poly (N-vinylimidazole) grafted chitosan as an effective adsorbent for mercury (II) removal from aqueous solution [37]. The same authors also assessed the used chitin-based materials in the study of the kinetic and thermodynamic of copper adsorption [147].

Those results open the path for future more environmentally friendly, biodegradable and low-cost water treatment solutions.

#### 2.3.4. Marine Biomaterials in Biomedical Applications

Marine biomaterials based on polysaccharides (chitin, fucoidan, alginate, etc.) and collagen are currently thoroughly investigated for the biomedical industry due to their biocompatibility and their specific properties in bone regeneration and wound healing [129,130]. The current studies focus mainly on three types of applications: tissue engineering, wound-dressing and drug delivery.

● Tissue Engineering

Whether it is for cartilage, bone or other cell tissue regeneration, high strength and low density porous scaffolds are required to both partially replace the structural and morphological roles of the damaged body part and also encourage and boost the regeneration process while providing enough vacancy for the ingrowth of cells.

It should be noted that natural marine biomaterials, such as sodium alginate [158] or collagen extracted from echinoderms [159], can be used unmodified as gels or membranes for guided tissue regeneration. Indeed, it has been shown that sodium alginate can be cross-linked with calcium to form gels 1 mm thick, able to support the proliferation of bone marrow cultures extracted from rats [158]. Successful results were also obtained for collagen membranes derived from echinoderms such as sea urchins and starfish. Furthermore, sea urchin collagen presents a significant advantage in term of eco-sustainability as it can be recycled from food wastes [159]. Furthermore, 3D chitosan scaffolds have an anti-inflammatory effect when used in biomaterials intended for wound healing applications [160].

However, the main drawback of the above biopolymers is their mechanical strength. However, this property can be enhanced by coupling different materials, hence the development of biocomposites.

In this regard, much research has been undertaken on chitosan, marine collagen- and alginate-based composites in particular. Mingxian Liu et al. showed that adding chitin nanocrystals as fillers inside a chitosan matrix results in a biocomposite scaffold with significantly greater compressive strength and modulus than the pure chitosan scaffold [161]. With its excellent biocompatibility and low cytotoxicity, this new type of scaffold promotes osteoblast cell adhesion and proliferation, making it an ideal candidate for bone tissue engineering.

Chitosan can also be combined with collagen and cellulose. Chitosan/collagen crosslinked hybrid scaffolds present better mechanical and degradation properties than their unitary systems separately, especially for the optimal ratio of 50/50 [162]. With good cytocompatibility, those scaffolds could be used for nerve tissue regeneration as they promote the attachment, migration and proliferation of Schwann cells [163]. Cellulose nanofibre-reinforced chitosan hydrogel is another example of biocomposite candidate for tissue engineering. Experiences done by Ingo Doench et al. show that such hydrogels could be used to restore damaged intervertebral disc tissue, approaching the functionality of a healthy disc [164].

Zubillaga et al. developed films and 3D porous scaffolds chitosan-genipin/chitin nanofillers as platforms for cartilage tissue engineering using adipose-derived mesenchymal stem cell chondrospheroids cultured in hypoxia [43,151].

Other fish collagen-based biocomposites also offer good opportunities to develop scaffold for 3D cell culture and cartilage regeneration [165,166]. For instance, research has been conducted on collagen scaffolds reinforced with HA, a non-sulphated polysaccharide that can be harvested from marine organisms, such as stingray livers and fish eyeballs [15,16]. The combination of those two biopolymers has produced a three-dimensional cross-linked structure with enhanced mechanical strength that has proved to improve adipose tissue development in vitro [167].

● Wound-Dressing

Beginning the early 2000s, work began on marine bio-sourced wound-dressing materials like chitin films [168]. With their flexibility, transparency, biodegradability, cytocompatibility and good adherence, the films have many advantages over existing commercial counterparts in terms of healing rate and efficiency. Following these initial findings, other biopolymer-based composites were developed. Biocompatible polymeric hydrogels such as polysaccharide and collagen films are often used as base material for effective wound-dressings as they promote rapid healing of the wound and do not cause secondary trauma to the new generated tissues on detachment.

For instance, alginate films loaded with asiaticoside (a substance extracted from the plant *Centella asiatica*) have shown promising results for usage as wound-dressings [169]. Biocomposite films made from gelatin (a collagen derivative) and chitin nanofibres have been designed for various medical applications, including wound-dressing [170]. Recently, it was shown that the addition of curcumin particles within a chitosan/collagen scaffold structure enhances cutaneous wound healing by regulating the complex interactions happening during the process [171].

● Drug Delivery

As stated above, polysaccharide-based biomaterials are an emerging class in numerous biomedical applications. In particular these materials have the key properties necessary for bio-sourced drug delivery systems: controllable biological activity, biodegradability, bacteriostatic properties and hydrogel-forming ability. Furthermore, using biodegradable polymers for controlled-delivery devices avoids the need for surgical removal after treatment.

Among those polymers, chitosan has been one of the most extensively studied for this application, especially since it can be used in as hydrogels, films, microspheres, nanoparticles, tablets, and so on [172,173]. For example, in 1994 Jameela and Jayakrishnan developed a drug delivery vehicle made from glutaraldehyde cross-linked chitosan microspheres [174]. With a slow release rate of the drug contained in the vehicle and good compatibility with living tissue, the material successfully sustains drug release over long periods.

Alginate is another marine polysaccharides with great potential for drug delivery materials, especially considering its mucoadhesives property, and is also often used in combination with chitosan. Indeed, chitosan/alginate microcapsules coated with PEG have already successfully been developed to encapsulate bioactive peptides (such as hirudin) design for oral delivery biomedication since 1998 [175]. In more recent years, chitosan/alginate nanoparticles blended in medical clay (Cloisite 30B) were used to develop bio-nanocomposites for controlled drug delivery as part of oral chemotherapy using curcumin as a prototype drug to inhibit tumor cells proliferation [176].

Marine-based biocomposites can also be used in other ways to enhance cancerous tumour treatments. For instance, porous chitosan/HA scaffolds have been designed to mimic the microenvironment of one of the most deadly brain tumour, glioblastoma multiforme [177]. Using the scaffold, 3D structures were created as similar as possible to the arrangement of cancer cells observed in brain tumours, in order to improve the accuracy of in vitro trials and help with cancer therapeutics screening.

This last example really shows the fascinating and promising combination between bio-sourcing and bio-mimicry approaches, which are both current trending topics in biomaterial development, especially regarding marine-based and marine-inspired materials.

## 3. The Ocean as a Source of Bio-Inspired Materials

Living in the oceans requires the ability to face important constraints regarding pressure, turbidity, continuity of the medium, dispersal by currents, salt corrosion, oxygen and nutrients in dissolved conditions, surface UV radiation, darkness and many others. Thanks to natural selection and evolution, marine organisms have developed a large spectrum of strategies to overcome these constraints. Many of these strategies are similar with industrial process requirements concerning filtration, fluid circulation, movement, adhesion, anti-adhesion, anticorrosion, communication and protection. In this context, numerous marine biological functions have been mimicked in the form of (bio)materials: respiration and nutrition (oxygen capture, nutrients filtration); locomotion (warp, fluids interactions); adhesion (mechanical bio-adhesives, biological glue, antifouling); protection (active molecules biosynthesis, composite materials, hierarchical structures); and communication (acoustic and chemical signals, inter/intra-species communication, camouflage). Examples of technological sectors that can take advantage of this approach are aeronautics, robotics, energy, medicine, pharmaceuticals, cosmetics, food industry, and information telecommunication.

In what follows, some examples of marine biomimetics and bio-inspired materials are described, regarding protection, adhesion and the modulation of light.

### 3.1. Mimicking the Functionalization of Reef Fish Mucus to Fight UVR-Induced Damages-Protection

High exposure to UVR is a threat to living systems because of the damage it causes to biological molecules. Historically, people have prevented skin- and eye-damage and disease by using protective clothing, hats, sunglasses and, more recently, cosmetic sunscreen products. Much progress on photoprotection has been made since para-aminobenzoic acid (PABA, one of the first available organic sunscreens ingredients) was originally patented in 1943 [178]. Today many synthetic organic and inorganic UV-proof products are available, but many have drawbacks including limited photostability and cross-stability. Furthermore, regular use of these products may have a harmful impact on human and ocean health [178,179,180,181]. Additionally, in certain circumstances, such as UV overexposure, and in certain risk groups, like patients suffering from *Albinism* and *Xeroderma pigmentosum,* current UV protection materials are insufficient leading to skin and ocular tissue injury. Therefore, novel efficient and environmentally friendly UV protection strategies, without these drawbacks, are being developed inspired by the strategies of marine organisms.

Certain marine organisms, namely cyanobacteria, corals, sponges, fungi and algae [182,183,184], have evolved an important biological strategy to deal with UV exposure. These organisms biosynthesize active molecules—MAAs, widely recognized as natural sunscreens. As mentioned before, the maximum absorption wavelengths (λ_max_) of MAAs range from 310 to 360 nm (UV-A and UV-B) with extraordinary high molar extinction coefficients (2800–50,000 M^−1^ cm^−1^) [184]. Interestingly, these molecules are accumulated in the external mucus and ocular lenses of reef fishes via bioaccumulation [185,186]. Typically in marine algae MAAs are free intracellular compounds, however, in some cyanobacteria, oligosaccharide-linked MAAs may also occur. In these compounds, MAAs are linked to oligosaccharide side chains leading to molecules that strongly interact with extracellular polysaccharides and proteins [184,187]. In shallow water fishes, MAAs are protein-associated and are present in the ocular tissues and also in the epidermis [184]. Inspired by this phenomenon, new biomaterials have been developed by mimicking this functionality of fish mucus [34].

Using green chemical reactions and exclusively marine natural compounds, MAAs were grafted to chitosan backbone and UV-absorbing solutions and films were made. The obtained functional biomaterial showed to be biocompatible, photoresistant and thermoresistant, and exhibit efficient absorption of both UV-A and UV-B radiations [34].

The advancement on the design of this kind of biomaterial could provide protection with a high efficiency over a wide range spectrum of UVR and avoiding the use of potentially deleterious compounds.

### 3.2. Adhesive Surfaces and Materials Inspired from Marine Organisms

Many organisms have developed surface adhesion strategies to protect themselves from predators and access to the resources needed for their growth. Other physiological functions, such as metamorphosis, molting and biomineralization, also involve adhesion mechanisms [188].

Bio-adhesion, specifically in wet environments, has received close attention and has been widely investigated for biomimetics. Biological adhesives offer impressive performances and are studied in the hopes of more reliable, efficient and environmentally-friendly glues [189]. In order to develop biologically-inspired adhesive surfaces and materials it is necessary both to understand the mechanisms used by the organisms and to characterize the biological adhesive comprehensively.

The wall-clinging free vertical and upside-down movement of geckos and tree frogs have been a great source of inspiration for the development of such products [190,191]. While the mechanisms involved there are physical [191] (van der Waals force [190], capillary force [192]) in the aquatic environment, attachment strategies developed by animals are predominantly chemical. They rely on highly viscous or solid adhesive secretions, usually containing specialized adhesive proteins [189]. By mimicking the use of said proteins, novel products can be developed for applications in dental, medical and industrial sectors for example [193,194,195].

Herein, two leading models of the biomimetic wet adhesion approach are briefly presented: the model used by marine mussels (Bivalvia) and the model used by sandcastle worms (Polychaeta: Sabellariidae).

#### 3.2.1. Marine Mussel Adhesion

To firmly adhere to different surfaces in marine ecosystems and stay attached under wave action and turbulence, mussels use adhesives unique to their class. Attachment is typically made via a byssus, a bundle of 50 to 100 adhesive threads and plaques secreted from a foot gland [196]. Once the target surface is selected, the mussel foot first adheres to the location, then secretes the byssus between the muscles within the shells and the attachment surface [194]. After complete adhesion, the foot generally retracts [197].

Byssus secretions are composed of polyphenolic proteins, collagen, and polyphenol oxidase - conferring water-resistant and high strength adhesion. In byssal threads, mussel proteins play a structural role to connect soft mussel tissues to adhesive plaques in the distal end. In the byssal plaques, proteins are responsible for mussel adhesion [188]. Among these proteins, six mussel foot proteins with substantially different sequences have been identified as the main actors of mussel adhesion. Each of them has a different location in the foot and byssus and each has a different role in the adhesion mechanism. Their mechanical and chemical properties have been studied for years and chemical synthesis of peptide analogues of mussel adhesion proteins is extensive [196,198,199].

These adhesive proteins contain a high quantity of post-translationally modified amino acids, such as 3,4-dihydroxyphenylalanine (DOPA) [200], which was shown to be essential for interaction between the mussel byssal threads and the surface [201]. The redox status of catechols in the DOPA side groups correlated with proteins interaction is a key parameter in mussel adhesion and must be considered for biomimetic applications. Indeed, reactive quinone species can be formed through the chemical or enzymatic oxidation of the catechol groups of DOPA. These species can further induce protein cross-linking via Michael addition and Schiff base formation with nucleophiles or radical aryl–aryl coupling with other catechols [202,203,204]. In the environment, mussels regulate pH to around 2 to ensure reducing conditions for adhesion establishment, pH then equilibrates to that of seawater once the foot disengages [205]. The strength of the attachment is conferred by different kinds of interactions comprising hydrogen bonding, metal–catechol coordination, electrostatic interaction, cation–π interaction and π–π aromatic interactions [197]. Environmental factors such as temperature, salinity, pH, nature of the substract and season are also determinant [200].

Thus, the catechol groups and the related DOPA chemistry have been extensively studied to inspire the development of adhesive materials for use in wet environments. In particular, due to their high biocompatibility, catechol-functionalized hydrogels incorporating DOPA or DOPA-containing short peptides, were developed for medical applications such as tissue adhesion, biomedical coatings, and drug delivery systems [200]. The coupling of DOPA amino acid residue to poly(ethylene glycol)(PEG) polymers is the main example but many other strategies have been developed and extensive details can be found in recent reviews on this field [197,200,203,206]. These hydrogels can display adhesive, self-healing, absorbing, antibacterial and antifouling properties [203]. For example, injectable adhesive hydrogels were developed that can be used as local haemostatic by forming adhesive barriers on bleeding sites and stop haemorrhages [207].

Catechol-based hydrogels can also be used for environmental applications as pollutant adsorbents in water purification [195]. Mussel adhesion strategies have also inspired other kind of applications including anti-corrosion coatings [208,209] and saltwater adhesives [210].

#### 3.2.2. Sandcastle Worms

*Phragmatopoma californica* (Polychaeta: Sabellariidae), commonly known as the sandcastle worm, is a marine tubeworm that builds underwater composite dwellings by gluing together sand grains and biomineral particles like shell fragments with a proteinaceous cement. At least four distinct secretory cell types from the worm building organ co-secrete products to form the glue [211]. The cement is able to adhere rapidly to various materials in seawater [212,213]. Stewart et al. proposed that the production of the glue by the worm follows a four steps mechanism, including a complex coacervation and condensation of adhesive proteins into secretory granules resulting in a closed cell foam structure of the glue [213]. The cement is produced in two parts containing highly acidic and basic proteins as well as divalent cations (calcium and magnesium). After secretion of the two substances, the glue sets very quickly through displacement of water from the mineral substrate inducing adhesion. The solidification of the cement by protein covalent cross-linking through oxidative coupling of DOPA similar to the one in mussel adhesion mechanism takes about six hours [213]. This results in a flexible but very tough structure strongly maintaining the biomineral particles together.

By mimicking the composition and curing mechanisms of the sandcastle worm cement, the team of Stewart et al. developed injectable biomedical adhesives targeting the healing of shattered bones. This novel biocompatible and biodegradable adhesive would allow for precise reconstruction of small bone fragments by maintaining their alignment without the use of conventional mechanical connectors like nails, pins and metal screws. The water-based adhesive, composed of synthetic water-soluble DOPA-containing copolymers and divalent cations, remains insoluble in wet environments and is able to bond to wet objects [214,215,216]. Researchers also took into consideration the processing mechanism of the glue used by the worm. As mentioned above, the oppositely-charged proteins forming the glue are stored in glands in individual granules preventing premature coacervation until secretion in the seawater. Using this approach, particles of a highly viscous polymer, poly(glycerol sebacate acrylate) (PGSA) were coated in alginate to reduce viscosity, resulting in an injectable aqueous dispersion that will crosslink only at the desired place and time by addition of positively-charged protamines and UV irradiation [217,218]. Commercial production and medical use of these kinds of bioinspired adhesives has begun. For example, after recent approval for clinical use in Europe, the company Gecko Biomedical is currently producing biomimetic sealants in the form of biomorphic and programmable synthetic polymers from PGSA for tissue reconstruction [217,219,220].

Several other medical adhesives inspired by marine coacervate formation have been developed and were recently reviewed elsewhere [194].

Many other marine organisms comprising octopus, barnacles, oysters and caddisfly larvae have developed adherence strategies and represent great sources of inspiration for the engineering of other adhesive materials [194,221,222,223,224]. Biomimetic approaches can also combine the mechanisms of several species at a same time. For instance, Cholewinski et al., recently developed Algae-mussel-inspired hydrogel composite resulting in a glue that does not require chemical conjugation, and can strongly bond dissimilar materials completely submerged in water [225]. Very recently, Han et al. were inspired by three different organisms (lotus leaf, mussel and sandcastle worm) to develop superhydrophobic surfaces with antithrombotic, antibiofouling, and tissue closure capabilities [226].

Adhesives thus represent a wide domain of marine-inspired functional materials, mainly in the biomedical sector but are also promising for (bio)technological, industrial, consumer and military applications and may soon become a part of everyday life [224,227].

### 3.3. Structural Colouration in the Marine Environment

The brilliant colours of marine organisms are the result of chemical and also physical properties. Much comes from inherently coloured molecules that absorb light (i.e., chlorophyll) or compounds that luminesce (i.e., luciferin). However, a myriad of deep structural colours (iridescence) in the marine environment come from the nanostructuration of the material and has evolved independently across many phyla [228], including crustaceans [229,230,231], fish [232,233,234,235], bivalves [236,237], cephalopods [238], algae [239,240,241,242], diatoms [243], bacteria [244], etc. In the animal world, this type of colouration has been linked to camouflage, predation, signal communication and sex choice [245]. Nonetheless, this section will focus on recent findings relating to light capture, focusing, and protection against UV [246] in the marine environment. The applications of structural colour materials for human technology are numerous and far from reaching full potential regarding optoelectronics, anti-counterfeiting, displays, sensors, soft robots, wearable electronics, organ-on-a-chip platforms and more [247].

Functional photonic architectures are structurally diverse in different species, and are mainly comprised in polymeric organic materials, including chitin, guanine, collagen, keratin, reflectin, pterin, melanin or carotenoids [248]. Silica, the inorganic principal component of the frustules (hard and porous cell wall or external layer of diatoms) is noteworthy exception in this regard.

Silvery fish use multilayers of guanine crystals which as act as an equivalent to a distributed Bragg reflector on their scales [235]. The natural reflectors predict a potential strategy for controllable micro-mirror structures [249], with possible use for nanomechanical sensing methods [250]. Iwasaka et al. demonstrated that individual bio-origin guanine crystals could be controlled under the influence of a magnetic field function as individual micromirrors. They showed reflectivity of the crystals could be switched off and on for controllable reflectivity [251]. Guanidine is not unique in forming organic reflectors: crystals of isoxanthopterin, a pteridine analog of guanine, were found to form both the image-forming “distal” mirror and the intensity-enhancing tapetum reflector in the compound eyes of some decapod crustaceans [252]. Other organic molecules could possibly be used to form crystals with superior reflective properties either in organisms or in artificial optical devices.

Among macroalgae, structural colouration is only reported limited genera of red and brown algae and with two mechanisms: extracellular multilayered structures [240] and iridescent bodies (*Dichotya* and *Cystoseira*), respectively. Structural colouration occurs in the red seaweeds *Chondrus chrispus*. It stems from extracellular lamellar structures comprising 1D photonic crystals. They are not ubiquitous across the surface of the organism or stage of lifecycle, only being present in the tips of the thalli of gametophytes, with maximum reflectance in the UV range [240], which points at a possible UV protective function. Similar structures were also found in the genre *Iridaea,* a species found principally in intertidal zones of the Pacific. The structures have not been characterized in detail, although are purported to be alternating layers of protein and polysaccharide-based materials [239]. The UV reflecting properties of this type of material was reproduced artificially using a biomimetic approach, although based on the structure butterfly wings (*Euploea mulciber*). Song et al. [253] used the wings as a template to deposit a Sn-gel coating, and the biological material was removed by calcination. The resulting material presented an additional UV reflection peak at about 350 nm, as compared to similar material without photonic structures.

The vivid iridescence of the brown seaweed *Cystoseira tamariscifolia* was recently discovered to be the result of intracellular opal-like lipid photonic crystals within photosynthetic cells [241]. The reflectance of the opaline vesicles is dynamically responsive to environmental illumination, being stronger in low light conditions than conditions of strong irradiance. The mechanism by which the algae produces the quasi-monodisperse lipids and controls the level of order or disorder, reversibly, remains unknown. The authors postulate that the biological function interacts with photosynthesis, scattering light efficiently to the chloroplasts in low-light conditions. This hypothesis is further supported by the habitat of the respective genera of iridescent brown algae species—below the mean low-tide lines, and being some of the deepest-growing species of seaweeds (especially *Dichotya* species). Photonic structures using these principles could be of use for the design of light-harvesting devices.

Giant clams of the genus *Tridacna* (Mollusca: Bivalvia) also display light-harvesting structural colouration. Their characteristic, vibrant colours come from the presence of specialized cells called iridocytes, under which are found high densities of photosynthetic symbiotic algae *Symbiodinium*. The iridocytes are distributed in such a manner photosynthetically productive wavelengths scattered laterally and forward and non-productive wavelengths are reflected backwards [236]. The spectral and angular scattering behaviour of an iridocyte depend on, and can be controlled by cell diameter, thicknesses of the high and low refractive index Bragg lamellae and their refractive indices [237]. Using these findings, Su et al. [254] designed synthetic iridocytes using silica nanoparticles in microspheres embedded in gelatin for use in optimal harvesting of solar energy. They used an emulsification-evaporation strategy to create spherical particles of tunable size in the micrometre scale, composed of individual silica particles of 200, 250, and 300 nm, which were suspended in gelatin-based films. The synthetic iridocytes showed wavelength selectivity, had little loss (with back-scattering intensity less than ≈0.01% of the forward-scattered intensity), and narrow forward scattering cone similar to that found in the organisms themselves.

In very recent findings, the colour of the eyes of the bay scallop *Argopecten irradians* (Mollusca: Bivalvia) were shown to stem from core-shell nanoparticles. Indeed the eyes contain a layer of close-packed non-mineralized nanospheres approximately 180 nm in diameter and consist of electron-dense cores approximately 140 nm in diameter surrounded by less electron-dense shells 20 nm thick. The authors used an optical modelling approach incorporating different refractive indexes of the core and the shell and found the nanospheres are an ideal size for producing angle-weighted scattering that is bright and blue. Important for the design of bio-inspired materials, this suggests that nanospheres with high-RI cores and low-RI shells produce brighter colours than nanospheres with a homogeneous RI; and that the colours of core–shell nanospheres may be tuned by altering the thickness of their low-RI shell [255].

Physical scattering of electromagnetic radiation also exists at sub-visible wavelengths: the silica-based external membranes, or frustules of diatoms have been shown to transform electromagnetic frequencies into the ultraviolet range [256,257,258]. Nanoscale features of frustules (holes, slits and ribs) are very reminiscent of photonic bandgap structures. These features may focus light at wavelengths close to photosynthetic optima [259], but also refract more potentially dangerous UVR wavelengths and convert them to blue light [258]. Layers of diatoms also present different transmission and reflectance of UVR, depending on characteristics of species-specific frustules [257].

Finally, photonic crystals are also shown to be used in conjunction with MAAs. In diatoms, the molecules were discovered imbedded in the silica matrix of the frustule, suggesting combinatory protective mechanisms [260]. This is not the only instance of interplay between MAAs and photonic structures in the marine environment—the eyes of the stomatopod species *Neogonodactylus oerstedii* where MAAs act as spectral tuning filters [231]. In addition to solar power applications for the conversion of non-optimal UV wavelengths to more efficient visible wavelengths. This natural architecture may inspire future work with nanoscale, non-photoreactive materials (as opposed to the widely used TiO_2_ and ZnO inorganic particles) for the design of UV resistant or protective materials.

## 4. Conclusions and Future Perspectives

The oceans are not just a source of raw materials but also inspiration for the design of novel materials. In this review, we have focused on both the use of biomolecules from organisms living in aquatic environments as biotechnological tools and the development of materials inspired on marine principles (i.e., by mimicking the functionality and structure of marine organisms), which can potentially solve current societal and environmental problems such as pollution, disposal of hazardous materials, and more. It focuses on the biodiversity of the marine environment and on the key properties of bioresources, residues and by-products as high added-value materials for achieving biomedical, pharmaceutical, cosmetic, food and nutrition and biotechnological applications.

Nonetheless, before use on a widespread scale, the durability, scalability, and real-world efficacy of the materials must be integrated into the development process. Many of these materials are proposed as ‘green’ or ‘eco-friendly’ alternatives to classic products. However, these credentials should be put to the test through testing that covers the entire lifecycle of the product—including discharge into the environment.

Biological resources are intrinsically renewable—coming from living things that grow and reproduce—if said resources are correctly managed. Many of the chitin, collagen, and algal-polysaccharide-based materials described above are by-products of extractive fisheries. Adding value to the harvest of natural resources can either incentivize respect of environmental regulation or encourage over-exploitation. Many of the functions described here are replicated using natural materials from other sources—however, destruction of habitats puts both the species and their esoteric survival strategies and structural adaptations at risk. Much progress is to be made in this highly trans-disciplinary field: the observations of field biologists are as valuable as the work of organic chemists and enzymologists that put the pieces together. As prominent as organic chemistry related to petrochemicals was in the past century, organic chemistry using biological materials may reveal itself to be just as important in the upcoming years of this century.

Taking a biomimetic approach requires in-depth knowledge of the intricacies of the living marine environment and green molecular extraction/modification to avoid the denaturation of the natural products involved. As such, for biomimetics to be valid philosophy for the design of novel materials, inter-disciplinarity appears to be a necessity, with expertise and feedback between experts in the fields of chemistry, structural mechanics, biochemistry, but also environmental chemistry and biology.

## Figures and Tables

**Figure 1 polymers-12-01002-f001:**
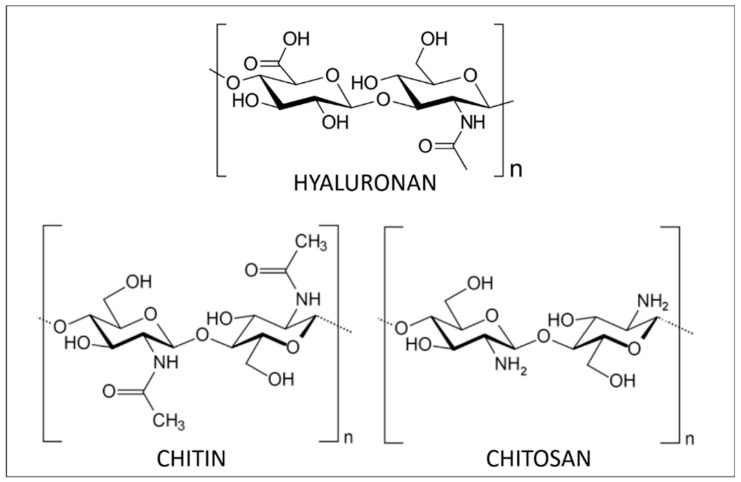
Chemical structure of marine polysaccharides: HA, chitin and chitosan.

**Figure 2 polymers-12-01002-f002:**
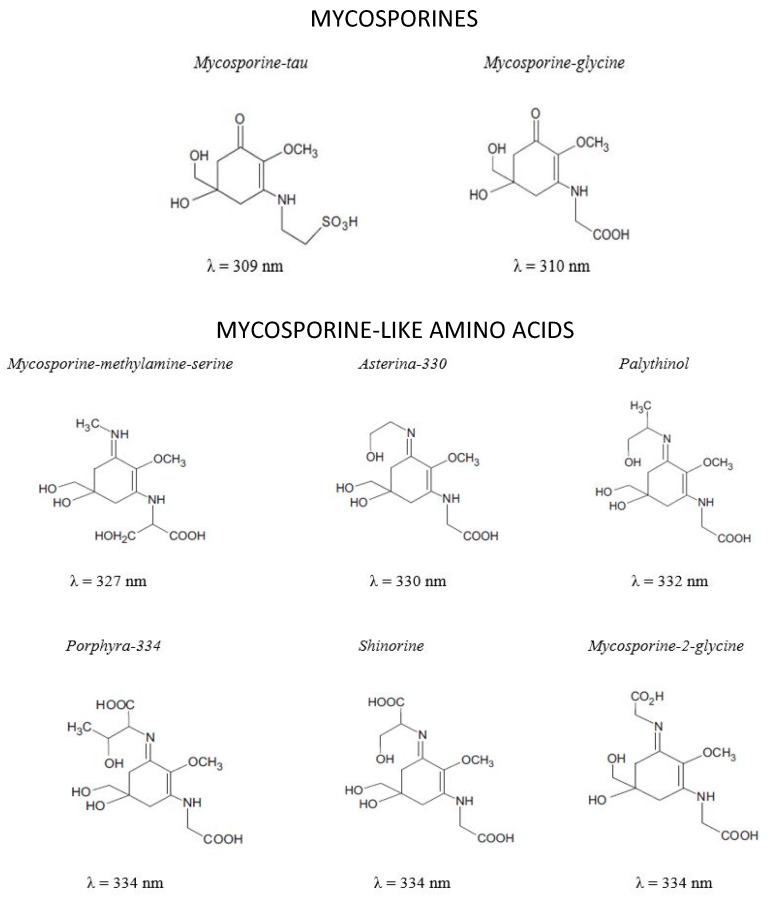
Identification, chemical structure and maximum absorption wavelengths (λmax) of different MAAs.

**Table 1 polymers-12-01002-t001:** Major cell wall and storage polysaccharides present in major macroalgal taxa [85,93].

Taxa	Crystalline	Hemicelluloses	Matrix Carboxylic	Matrix-Sulfated	Storage
**Chlorophyceae**	Cellulose	Xyloglucan Mannans Glucuronan (1-3) β-glucan	Ulvans	Ulvans	Inlulin (fructan) Laminaran Strach
**Rhodophytae**	Cellulose (1-4)-β-d-mannans (1-4)-β-d-xylans (1-3)-β-d-xylan Chitin	Glucomannan Sulfated mixed-linkage glucan (1-3)(1-4)-β-d-xylan	-	Agars Carrageenans Porphyran Mannans	Floridean glycogen
**Phaeophyceae**	Cellulose	Sulfated xylofucoglucan Sulfated xylofuco-glucouronan (1-3)-β-glucan	Alginates	Homofucans	Laminaran

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
