# Peer review of "Marine-Derived Polymeric Materials and Biomimetics: An Overview"

_polymers, 2020, doi:10.3390/polym12051002_

Round 1

Reviewer 1 Report

The authors have described very interesting and actual scientific theme concerning marine-derived polymeric materials for their potential biotechnological application. The manuscript is clearly described and well presented. I have only  two minor aspects for corrections:

  1. A resolution quality of Figure 1 should be improved.
  2. Table 1: width of the three last columns should be enlarged to improve given description of these columns

Author Response

(in black, reviewer’s comment; in blue our reply)

Reviewer 1

The authors have described very interesting and actual scientific theme concerning marine-derived polymeric materials for their potential biotechnological application. The manuscript is clearly described and well presented. I have only  two minor aspects for corrections:

We would like to thank Reviewer 1 for the positive comments on the relevance of our work.

  1. A resolution quality of Figure 1 should be improved.

We agree with the reviewer comment. We have now improved the quality of Figure 1. A new figure was incorporated in the revised manuscript.

  1. Table 1: width of the three last columns should be enlarged to improve given description of these columns

The three last columns were enlarged as suggested by the reviewer.

Reviewer 2 Report

Type of manuscript: Review
Title: Marine-Derived Polymeric Materials and Biomimetics: an Overview
Authors: Marion Claverie, Colin McReynolds, Arnaud Petitpas, Martin Thomas,
Susana C.M. Fernandes *
Submitted to section: Biobased and Biodegradable Polymers

The above mentioned manuscript, presents an overview of some important recent marine-derived polymeric materials.
The manuscript is well-organized and written, I am positive with its publication.
Only o small concern about the figures clarity and significance.

Author Response

(in black, reviewer’s comment; in blue our reply)

Reviewer 2

The above mentioned manuscript, presents an overview of some important recent marine-derived polymeric materials.

The manuscript is well-organized and written, I am positive with its publication.

Also, we would like to thank Reviewer 2 for the positive comments on the relevance of our work.

Only o small concern about the figures clarity and significance.

We agree with the reviewer concerning the significance of this figure. We have now improved the quality of the figures and Figure 3 was removed.

Reviewer 3 Report

Dear Editor,

in the present review the Marine-Derived Polymeric Materials and some of their applications have been extensively described. The review is well organized and summarizes adequately these biopolymers and last findings. For this reason I propose to accept for publication. I have some proposals, mainly in Chitosan and its derivatives that I have high experience, and I hope to find interesting for authors.

It was mentioned that chitosan can be modified in order to have better properties. Such modified chitosan derivatives have been extensively described in a recent review, which I think that should be added in the particular session. Please see: Kyzas G., Bikiaris D. Recent modifications of chitosan for adsorption applications: A critical and systematic review. Mar. Drugs 13; 312-337: 2015.

The applications of Chitosan and its derivatives, as they are presented, reduce and minimise its important. There are any applications of chitosan as nanoparticles for drug delivery applications and mainly for ocular drug release. Please see.

Nanoparticles for drug release

Koukaras E.N., Papadimitriou S.A, Bikiaris D.N., Froudakis G.E. Insight on the formation of chitosan nanoparticles through ionotropic gelation with tripolyphosphate. Mol. Pharm. 9; 2856−2862: 2012.

Koukaras E.N., Papadimitriou S.A., Bikiaris D.N., Froudakis G.E. Properties and energetics for design and characterization of chitosan nanoparticles used for drug encapsulation. RSC Adv., 4; 12653-12661: 2014.

Occular release formulations

Papadimitriou S., Bikiaris D., Avgoustakis K., Karavas E., Georgarakis M. Chitosan nanoparticles loaded with dorzolamide and pramipexole. Carbohyd. Polym. 73; 44-54: 2008.

Siafaka P.I., Titopoulou A., Koukaras E.N., Kostoglou M., Koutris E., Karavas E., Bikiaris D.N. Chitosan derivatives as effective nanocarriers for ocular release of timolol drug. Int. J. Pharm. 495; 249–264: 2015.

Also, chitosan is an ideal material for Controlled release formulations. Please see:

Koutroumanis K.P., Avgoustakis K., Bikiaris D. Synthesis of cross-linked N-(2-carboxybenzyl)chitosan pH sensitive polyelectrolyte and its use for drug controlled delivery Carbohydr. Polym. 82; 181-188: 2010.

Nanaki S., Tseklima M., Christodoulou E., Triantafyllidis K., Kostoglou M., Bikiaris D.N. Thiolated Chitosan Masked Polymeric Microspheres with Incorporated Mesocellular Silica Foam (MCF) for Intranasal Delivery of Paliperidone. Polymers 9; 617 (1-21); doi:10.3390/polym9110617. 2017.

Lazaridou M., Christodoulou E., Nerantzaki M., Kostoglou M., Lambropoulou D.A., Katsarou A., Pantopoulos K., Bikiaris D.N. Formulation and In-Vitro Characterization of Chitosan-Nanoparticles Loaded with the Iron Chelator Deferoxamine Mesylate (DFO). Pharmaceutics 12, 238, 2020.

Concerning the biomedical applications of chitosan and its derivatives there are several of them with appropriate drugs or with appropriate structures for wood dressing and scaffold applications. Please see:

Wood dressing

Siafaka P.I., Zisi A., Exindari M., Karantas I.D., Bikiaris D.N. Porous dressings of modified Chitosan with poly(2-hydroxyethyl acrylate) for topical wound delivery of Levofloxacin. Carboh. Polym. 143; 90-99: 2016.

Michailidou G., Christodoulou E., Nanaki S., Barmpalexis P., Karavas E., Vergkizi-Nikolakaki S., Bikiaris D.N. Super-hydrophilic and high strength polymeric foam dressings of modified chitosan blends for topical wound delivery of chloramphenicol. Carbohydrate Polymers 208; 1-13: 2019.

Scaffolds

Nerantzaki M.C., Koliakou I.G., Kaloyianni M.G., Terzopoulou Z.N., Siska E.K., Karakassides M.A., Boccaccini A.R., Bikiaris D.N. New N-(2-carboxybenzyl)chitosan composite scaffolds containing nanoTiO2 or Bioglass for Bone-tissue engineering applications. Int. J. Polymeric Mater. Polymeric Biomater. 66; 71-81: 2017.

Finally, chitosan id also used as appropriate polymer to remove several toxic contaminates, and this application is not discussed in the review. Please see some important findings:  

Dyes removal

Lazaridis N.K., Kyzas G.Z., Vassiliou A.A., Bikiaris D.N. Chitosan Derivatives as Biosorbents for Basic Dyes. Langmuir 23; 7634-7643: 2007.

Kyzas G.Z., Bikiaris D.N., Lazaridis N.K. Low-Swelling Chitosan Derivatives as Biosorbents for Basic Dyes. Langmuir 24; 4791-4799: 2008.

Kyzas G.Z., Bikiaris D.N., Mitropoulos A.C. Chitosan adsorbents for dye removal: a review. Polym. Int. 66; 1800–1811: 2017.

Toxic metal removal

Kyzas G.Z., Kostoglou M., Lazaridis N.K., Bikiaris D.N. N-(2-carboxybenzyl) grafted chitosan as adsorptive agent for simultaneous removal of positively and negatively charged toxic metal ions. J. Hazard. Mater. 244-245; 29-38: 2013.

Kyzas G.Z., Siafaka P.I., Lambropoulou D.A., Lazaridis N.K., Bikiaris D.N. Poly(itaconic acid)-grafted chitosan adsorbents with different cross–linking for Pb(II) and Cd(II) uptake. Langmuir, 30; 120-131: 2014.

Kyzas G.Z., Siafaka P.I., Pavlidou E.G., Chrissafis K.J., Bikiaris D.N. Synthesis and adsorption application of succinyl-grafted chitosan for the simultaneous removal of zinc and cationic dye from binary hazardous mixtures.  Chem. Eng. J. 259; 438-448: 2015.

Pharmaceutical contaminates removal

Kyzas G.Z., Kostoglou M., Lazaridis N.K., Lambropoulou D., Bikiaris D.N. Environmental friendly technology for the removal of pharmaceutical contaminants from wastewaters using modified chitosan adsorbents. Chem. Eng. J. 222; 248-258: 2013.

Kyzas G.Z., Bikiaris D.N., Lambropoulou D.A. Effect of humic acid on pharmaceuticals adsorption using sulfonic acid grafted chitosan. J. Molecul. Liquids 230; 1–5: 2017.

Tzereme A., Christodoulou E., Kyzas G.Z., Kostoglou M., Bikiaris D.N., Lambropoulou D.A. Chitosan Grafted Adsorbents for Diclofenac Pharmaceutical Compound Removal from Single-Component Aqueous Solutions and Mixtures. Polymers 11, 497; 2019.

Author Response

(in black, reviewer’s comment; in blue our reply)

Reviewer 3

In the present review the Marine-Derived Polymeric Materials and some of their applications have been extensively described. The review is well organized and summarizes adequately these biopolymers and last findings. For this reason I propose to accept for publication. I have some proposals, mainly in Chitosan and its derivatives that I have high experience, and I hope to find interesting for authors.

Again, we would like to thank Reviewer 3 for the positive comments on the relevance of our work.

It was mentioned that chitosan can be modified in order to have better properties. Such modified chitosan derivatives have been extensively described in a recent review, which I think that should be added in the particular session. Please see: Kyzas G., Bikiaris D. Recent modifications of chitosan for adsorption applications: A critical and systematic review. Mar. Drugs 13; 312-337: 2015.

The applications of Chitosan and its derivatives, as they are presented, reduce and minimise its important. There are any applications of chitosan as nanoparticles for drug delivery applications and mainly for ocular drug release. Please see.

Nanoparticles for drug release 

Koukaras E.N., Papadimitriou S.A, Bikiaris D.N., Froudakis G.E. Insight on the formation of chitosan nanoparticles through ionotropic gelation with tripolyphosphate. Mol. Pharm. 9; 2856−2862: 2012.

Koukaras E.N., Papadimitriou S.A., Bikiaris D.N., Froudakis G.E. Properties and energetics for design and characterization of chitosan nanoparticles used for drug encapsulation. RSC Adv., 4; 12653-12661: 2014.

Occular release formulations

Papadimitriou S., Bikiaris D., Avgoustakis K., Karavas E., Georgarakis M. Chitosan nanoparticles loaded with dorzolamide and pramipexole. Carbohyd. Polym. 73; 44-54: 2008.

Siafaka P.I., Titopoulou A., Koukaras E.N., Kostoglou M., Koutris E., Karavas E., Bikiaris D.N. Chitosan derivatives as effective nanocarriers for ocular release of timolol drug. Int. J. Pharm. 495; 249–264: 2015.

Also, chitosan is an ideal material for Controlled release formulations. Please see:

Koutroumanis K.P., Avgoustakis K., Bikiaris D. Synthesis of cross-linked N-(2-carboxybenzyl)chitosan pH sensitive polyelectrolyte and its use for drug controlled delivery Carbohydr. Polym. 82; 181-188: 2010.

Nanaki S., Tseklima M., Christodoulou E., Triantafyllidis K., Kostoglou M., Bikiaris D.N. Thiolated Chitosan Masked Polymeric Microspheres with Incorporated Mesocellular Silica Foam (MCF) for Intranasal Delivery of Paliperidone. Polymers 9; 617 (1-21); doi:10.3390/polym9110617. 2017.

Lazaridou M., Christodoulou E., Nerantzaki M., Kostoglou M., Lambropoulou D.A., Katsarou A., Pantopoulos K., Bikiaris D.N. Formulation and In-Vitro Characterization of Chitosan-Nanoparticles Loaded with the Iron Chelator Deferoxamine Mesylate (DFO). Pharmaceutics 12, 238, 2020.

Concerning the biomedical applications of chitosan and its derivatives there are several of them with appropriate drugs or with appropriate structures for wood dressing and scaffold applications. Please see:

Wood dressing

Siafaka P.I., Zisi A., Exindari M., Karantas I.D., Bikiaris D.N. Porous dressings of modified Chitosan with poly(2-hydroxyethyl acrylate) for topical wound delivery of Levofloxacin. Carboh. Polym. 143; 90-99: 2016.

Michailidou G., Christodoulou E., Nanaki S., Barmpalexis P., Karavas E., Vergkizi-Nikolakaki S., Bikiaris D.N. Super-hydrophilic and high strength polymeric foam dressings of modified chitosan blends for topical wound delivery of chloramphenicol. Carbohydrate Polymers 208; 1-13: 2019.

Scaffolds

Nerantzaki M.C., Koliakou I.G., Kaloyianni M.G., Terzopoulou Z.N., Siska E.K., Karakassides M.A., Boccaccini A.R., Bikiaris D.N. New N-(2-carboxybenzyl)chitosan composite scaffolds containing nanoTiOor Bioglass for Bone-tissue engineering applications. Int. J. Polymeric Mater. Polymeric Biomater. 66; 71-81: 2017.

Finally, chitosan id also used as appropriate polymer to remove several toxic contaminates, and this application is not discussed in the review. Please see some important findings:  

Dyes removal

Lazaridis N.K., Kyzas G.Z., Vassiliou A.A., Bikiaris D.N. Chitosan Derivatives as Biosorbents for Basic Dyes. Langmuir 23; 7634-7643: 2007.

Kyzas G.Z., Bikiaris D.N., Lazaridis N.K. Low-Swelling Chitosan Derivatives as Biosorbents for Basic Dyes. Langmuir 24; 4791-4799: 2008.

Kyzas G.Z., Bikiaris D.N., Mitropoulos A.C. Chitosan adsorbents for dye removal: a review. Polym. Int. 66; 1800–1811: 2017.

Toxic metal removal

Kyzas G.Z., Kostoglou M., Lazaridis N.K., Bikiaris D.N. N-(2-carboxybenzyl) grafted chitosan as adsorptive agent for simultaneous removal of positively and negatively charged toxic metal ions. J. Hazard. Mater. 244-245; 29-38: 2013.

Kyzas G.Z., Siafaka P.I., Lambropoulou D.A., Lazaridis N.K., Bikiaris D.N. Poly(itaconic acid)-grafted chitosan adsorbents with different cross–linking for Pb(II) and Cd(II) uptake. Langmuir, 30; 120-131: 2014.

Kyzas G.Z., Siafaka P.I., Pavlidou E.G., Chrissafis K.J., Bikiaris D.N. Synthesis and adsorption application of succinyl-grafted chitosan for the simultaneous removal of zinc and cationic dye from binary hazardous mixtures.  Chem. Eng. J. 259; 438-448: 2015.

Pharmaceutical contaminates removal 

Kyzas G.Z., Kostoglou M., Lazaridis N.K., Lambropoulou D., Bikiaris D.N. Environmental friendly technology for the removal of pharmaceutical contaminants from wastewaters using modified chitosan adsorbents. Chem. Eng. J. 222; 248-258: 2013.

Kyzas G.Z., Bikiaris D.N., Lambropoulou D.A. Effect of humic acid on pharmaceuticals adsorption using sulfonic acid grafted chitosan. J. Molecul. Liquids 230; 1–5: 2017.

Tzereme A., Christodoulou E., Kyzas G.Z., Kostoglou M., Bikiaris D.N., Lambropoulou D.A. Chitosan Grafted Adsorbents for Diclofenac Pharmaceutical Compound Removal from Single-Component Aqueous Solutions and Mixtures. Polymers 11, 497; 2019.

We have now selected and incorporated some of the references suggested by the reviewer. To do so, we took into account the most recent references (last 5 years) of the topics described in the manuscript and we also add two applications suggested by the reviewer: dyes removal and pharmaceutical contaminates removal.

Below the list of selected references:

Kyzas G., Bikiaris D. Recent modifications of chitosan for adsorption applications: A critical and systematic review. Mar. Drugs 13; 312-337: 2015.

Koukaras E.N., Papadimitriou S.A., Bikiaris D.N., Froudakis G.E. Properties and energetics for design and characterization of chitosan nanoparticles used for drug encapsulation. RSC Adv., 4; 12653-12661: 2014.

Siafaka P.I., Titopoulou A., Koukaras E.N., Kostoglou M., Koutris E., Karavas E., Bikiaris D.N. Chitosan derivatives as effective nanocarriers for ocular release of timolol drug. Int. J. Pharm. 495; 249–264: 2015.

Lazaridou M., Christodoulou E., Nerantzaki M., Kostoglou M., Lambropoulou D.A., Katsarou A., Pantopoulos K., Bikiaris D.N. Formulation and In-Vitro Characterization of Chitosan-Nanoparticles Loaded with the Iron Chelator Deferoxamine Mesylate (DFO). Pharmaceutics 12, 238, 2020.

Siafaka P.I., Zisi A., Exindari M., Karantas I.D., Bikiaris D.N. Porous dressings of modified Chitosan with poly(2-hydroxyethyl acrylate) for topical wound delivery of Levofloxacin. Carboh. Polym. 143; 90-99: 2016.

Michailidou G., Christodoulou E., Nanaki S., Barmpalexis P., Karavas E., Vergkizi-Nikolakaki S., Bikiaris D.N. Super-hydrophilic and high strength polymeric foam dressings of modified chitosan blends for topical wound delivery of chloramphenicol. Carbohydrate Polymers 208; 1-13: 2019.

Nerantzaki M.C., Koliakou I.G., Kaloyianni M.G., Terzopoulou Z.N., Siska E.K., Karakassides M.A., Boccaccini A.R., Bikiaris D.N. New N-(2-carboxybenzyl)chitosan composite scaffolds containing nanoTiOor Bioglass for Bone-tissue engineering applications. Int. J. Polymeric Mater. Polymeric Biomater. 66; 71-81: 2017.

Kyzas G.Z., Bikiaris D.N., Mitropoulos A.C. Chitosan adsorbents for dye removal: a review. Polym. Int. 66; 1800–1811: 2017.

Kyzas G.Z., Siafaka P.I., Pavlidou E.G., Chrissafis K.J., Bikiaris D.N. Synthesis and adsorption application of succinyl-grafted chitosan for the simultaneous removal of zinc and cationic dye from binary hazardous mixtures.  Chem. Eng. J. 259; 438-448: 2015.

Tzereme A., Christodoulou E., Kyzas G.Z., Kostoglou M., Bikiaris D.N., Lambropoulou D.A. Chitosan Grafted Adsorbents for Diclofenac Pharmaceutical Compound Removal from Single-Component Aqueous Solutions and Mixtures. Polymers 11, 497; 2019.